# Recurrent Episodes of Some Mosquito-Borne Viral Diseases in Nigeria: A Systematic Review and Meta-Analysis

**DOI:** 10.3390/pathogens11101162

**Published:** 2022-10-08

**Authors:** Anyebe Bernard Onoja, Arome Cornelius Omatola, Mamoudou Maiga, Ishaya Samuel Gadzama

**Affiliations:** 1Department of Virology, College of Medicine, University of Ibadan, Ibadan 200284, Nigeria; 2Department of Microbiology, Kogi State University, Anyigba 272102, Nigeria; 3Center for Innovation in Global Health Technologies, Evanston Campus, Northwestern University, Evanston, IL 60202, USA; 4Department of Microbiology, Taraba State University, Jalingo 660213, Nigeria

**Keywords:** *Aedes* species, arbovirus, dengue, Chikungunya, yellow fever virus, Nigeria

## Abstract

Different ecological zones favor the breeding of *Aedes* species. The molecular epidemiology of dengue virus (DENV), yellow fever virus (YFV), and Chikungunya virus (CHIKV) was determined from outbreaks and surveillance activities in Nigeria. Twenty-eight DENV, twenty-five YFV, and two CHIKV sequences from Nigeria were retrieved from GenBank. Genotyping was performed with a genome detective typing tool. The evolutionary comparison was performed by the Maximum Likelihood method on MEGA. Chi-square was used to compare the association between the proportions of viral infections at different times. Six DENV-1 were detected in 1964, 1965, 1978, 2007, and 2018. Nineteen DENV-2 strains were reported, four belonging to sylvatic VI, one belonging to cosmopolitan II, and twelve to Asian I genotype V. DENV-2 genotype VI was detected in 1966, and genotypes II and V in 2019. All three DENV-3 were detected in 2018, while only one DENV-4 was identified in 2019. YFV was reported in 1946 and then in the 60s, 70s, 80s, 90s, 2018, and 2019 with reports to date. CHIKV is still circulating following its identification in 1964 and 1965. Recurrent episodes of dengue, Chikungunya, and yellow fever continue unabated. Vector control initiatives and immunization should be greatly sustained.

## 1. Introduction

There are over seventy members of the genus Flavivirus, the family Flaviviridae, of which thirteen are a threat to public health. Two of these, yellow fever virus and dengue virus, along with Chikungunya which is an alphavirus, rank among the most important arthropod-borne diseases in the developing world. They are transmitted by mosquitoes which have been reported to spread across international boundaries, thereby facilitating spread. *Aedes (St.) aegypti* is the principal vector in most locations [1]. Yellow fever epidemics have been reported in Angola, the Democratic Republic of Congo, Sudan, Nigeria, and across Africa [2,3]. Similarly, dengue has emerged as another important arthropod-borne viral disease in humans in the last 40 years. The frequency of dengue fever epidemics has increased dramatically; hyperendemic transmission has been established over a geographically expanding range [4]. Twenty-seven African countries have been identified by the European Network on Imported Infectious Disease Surveillance to be sources where travelers acquire mosquito-borne infections [5]. About 80–100 million annual dengue infections occurred worldwide based on an assumed constant annual infection rate. Globally, it is now estimated that about 390 million cases of dengue fever occur annually [4]. Of great concern is the emergence of DHF in Africa in recent years [6]. 

Dengue has been reported in most African countries and the vector, *Ae. Aegypti*, is found throughout the continent, except in the Maghreb states because of their arid climate [7]. Until 2010, little was known about dengue in Tanzania. Since then, four outbreaks have been reported in Dar es Salaam City, Tanzania [8]. Outbreaks and clinical cases of dengue fever were reported from Port Sudan at the Red Sea in the north-east [9], and in Kassala, Kordofan, and Darfur in Sudan [10]. Simultaneous Chikungunya and dengue outbreaks have been reported in Gabon [11] and Sierra Leone [12]. A resurgence of Chikungunya epidemics was reported in the DRC [13]. 

After the initial reports in Ibadan in 1964, CHIK reappeared in epidemic form, with virus isolations from man and mosquitoes. A decade later, the second epidemic of Chikungunya occurred among children in Ibadan, Nigeria in 1974 [14]. Since then, it has expanded to other areas in the rainforest region with reports in the Guinea Savannah region [15]. There are reports of overseas exports of mosquito-borne viruses by travelers visiting Nigeria [16]. Changes in human ecology and vector behavior are factors that are responsible for an increase in the incidence and distribution of mosquito-borne diseases [17]. These changes have been accompanied by an unwillingness to undertake effective mosquito control [18]. Rapid globalization, together with the presence of *Aedes* species, reports of arbovirus cases from travelers, and seroprevalence surveys, indicate that West Africa is the emerging front for arbovirus surveillance and control [19]. 

The goal of this study is to determine the evolutionary diversity of dengue, yellow fever, and Chikungunya virus to understand the epidemiology and appraise the recurrence of these diseases in Nigeria.

## 2. Materials and Methods 

Study Design: In this study, we retrieved all dengue virus sequences deposited in the National Centre for Biotechnology Information (NCBI) GenBank from Nigeria from 1966 to 2019, 28 yellow fever virus isolates from 1946 to 2019 and 2 Chikungunya virus sequences from 1964–1965. The search words in the nucleotides and genome database were dengue virus, yellow fever virus, Chikungunya, and Nigeria. These three viruses, deposited at different times from different parts of the world, were retrieved for evolutionary comparisons. 

Phylogenetic Analysis: The evolutionary history was inferred using the Maximum Likelihood method based on the Tamura 3-parameter model [20]. The percentage of trees in which the associated taxa clustered together is shown next to the branches. Trees were obtained by applying neighbor-joining [21] and BioNJ algorithms to matrixes of pairwise distances estimated using the Maximum Composite Likelihood (MCL) approach [22], and selecting the topology with a superior log likelihood value. The robustness of phylogenetic groupings was assessed with 1000 replicates of bootstrap values [23]. The trees were drawn to scale, with branch lengths measured in the number of substitutions per site. There were 37 DENV1 sequences used for phylogenetic analysis; 30 sequences were obtained from Latin America, Asia, and Europe, and seven sequences from Nigeria. Fifty-two DENV-2 sequences were used for the phylogenetic analysis, with nineteen from Nigeria and thirty-three from Asia, Latin America, New Guinea, and other African countries. There were 22 DENV-4 sequences used for the phylogenetic diversity, with a 2019 sequence from Nigeria. Fifty-six YFV sequences from the Genbank were used for the phylogenetic analysis, with twenty-five sequences from 1946 to 2019 in Nigeria. A total of 32 CHIKV isolates were used in this study from other African countries (two from Nigeria), Asia, India, Latin America, the USA, and Europe. Evolutionary analyses were conducted in MEGA 7 [24]. 

Genotyping and Ecotype Determination: Genotyping of DENV and YFV isolates from Nigeria was carried out using the online version 1.130 available at https://www.genomedetective.com/app/typingtool/virus/ (accessed on 6 May 2021). The YFV isolates were selected based on four genotypes around the world, representing West African, East African, South American I, and South American II.

Statistical Analysis: Statistical Packages for Social Sciences, SPSS 22 software (IBM Corp. released 2013, IBM SPSS Statistics for Windows, version 20.0, Armonk, NY, USA) was used to analyze data obtained in the study. Chi-square was used to compare the association between proportions of infections at different times in various locations. *p* ≤ 0.05 was considered significant.

## 3. Results

Dengue is highly endemic in Nigeria, with intermittent outbreaks. DENV-1 was identified in 1964 in Ibadan, with subsequent reports in 1965 and 1978, then in 2018 in Lagos. All of the DENV-2 identified in humans in 1966 were sylvatic strains which changed in 2018 with the Cosmopolitan strain identified in 2019 in Saki, Oyo State, and a major dengue outbreak caused by the Asian I lineage in Edo State, Nigeria (Table 1). All the DENV serotypes were identified from 1964 to 2019. DENV-2 accounts for 66% of the molecular evidence of dengue in the GenBank from Nigeria, with few reports of sylvatic dengue. This is followed by DENV-1 (21%), DENV-3 (10%), and DENV-4 (3%). DENV-2, belonging to genotype V, predominated over the other genotype constellations, and the difference in human seropositive status of DENV between genotype V and other circulating DENV genotypes is statistically significant (*p* = 0.01). The Asian type, with a prevalence of 41%, appeared to be more relevant epidemiologically in comparison with the other strains (*p* = 0.01). Generally, there was an increasing trend in reported cases of DENV in recent years. In 2019, about 48% of DENV infections were reported, a rate that was 2- and 16-fold higher than 24% and 3% in 2018 and 2007, respectively. In terms of the geographical distribution of DENV infections, significantly more cases of infections were reported in Benin City (south-south Nigeria) where DENV-2 genotype V predominated compared to the other locations (*p* = 0.02). This was followed by unassigned genotypes of DENV-1 and DENV-3 (Table 2). 

Seven DENV-1 sequences were identified from Nigeria. DENV-1 sequences detected in 1968 and 1978 in Ibadan clustered with a 2019 Brazil sequence. A separate cluster obtained in 2018 from Lagos showed closer ancestry to sequences from Brazil in 2007 and 2010 and from the USA in 2010, as well as from India in 2015. Further, a sequence from the DRC in 2015 was closely related to a historical sequence in Nigeria (Figure 1). DENV-2 was more predominant, with 19 sequences from Nigeria. The thirteen most recent sequences formed a unique cluster with a 1996 sequence responsible for DHF and another 1974 sequence from Thailand. These were closely related to other sequences from China, Jamaica, and the 1944 reference DENV-2 strain from New Guinea (Figure 2). The cosmopolitan strain from Saki, Nigeria is closely related to a sequence from Burkina Faso in 1983. In the last node of the phylogenetic tree, apart from the 2018 sequence from Lagos which stands alone, the four sequences identified in 1966 from humans clustered with sylvatic strains from mosquitoes in Senegal, Guinea, Burkina Faso, and Cote d’Ivoire (Figure 2). The DENV-4 sequence in Nigeria is closely related to a Brazilian sequence identified in 2014 (Figure 3).

Yellow fever outbreaks occurred in 1970 and 1987, with a spike in 2019 in Nigeria (Figure 4). Two CHIKV sequences from Nigeria were identified in 1964 and 1965 in Ibadan. Both sequences are closely related to strains from Indonesia in 2018, the Philippines in 2013, the USA in 2014, and Nicaragua in 2015 (Figure 5).

All DENVs obtained were from humans and can be retrieved from the GenBank with the accession numbers below. Several DENVs detected in Nigeria could not be assigned genotypes and ecotypes. More reports of Asian I lineage were from Benin in 2019, followed by Lagos and Ibadan. Sylvatic strains are given way to different ecotypes.

Detection was from Ibadan and Saki in Oyo State and Lagos State in southwest Nigeria, and Benin in Edo State, south-south Nigeria. All four DENV serotypes were reported with a change from sylvatic to Asian ecotypes. More DENV-2 genotype V Asian lineage was detected in 2019 from Benin in Edo State.

The evolutionary history was inferred by using the Maximum Likelihood method based on the Tamura 3-parameter model [20]. The tree with the highest log likelihood (−16,908.06) is shown. The percentage of trees in which the associated taxa clustered together is shown next to the branches. Initial tree(s) for the heuristic search were obtained automatically by applying neighbor-join and BioNJ algorithms to a matrix of pairwise distances estimated using the Maximum Composite Likelihood (MCL) approach, and then selecting the topology with superior log likelihood value. The tree is drawn to scale, with branch lengths measured in the number of substitutions per site. The analysis involved 37 nucleotide sequences. Codon positions included were 1st + 2nd + 3rd + Noncoding. There were a total of 1487 positions in the final dataset. Evolutionary analyses were conducted in MEGA7 [24].

The evolutionary history was inferred by using the Maximum Likelihood method based on the Tamura 3-parameter model [20]. The tree with the highest log likelihood (−11,464.48) is shown. The percentage of trees in which the associated taxa clustered together is shown next to the branches. Initial tree(s) for the heuristic search were obtained automatically by applying neighbor-join and BioNJ algorithms to a matrix of pairwise distances estimated using the Maximum Composite Likelihood (MCL) approach, and then selecting the topology with superior log likelihood value. The tree is drawn to scale, with branch lengths measured in the number of substitutions per site. The analysis involved 52 nucleotide sequences. Codon positions included were 1st + 2nd + 3rd + Noncoding. There were a total of 1245 positions in the final dataset. Evolutionary analyses were conducted in MEGA7 [24].

The evolutionary history was inferred by using the Maximum Likelihood method based on the Tamura 3-parameter model [20]. The tree with the highest log likelihood (−9065.87) is shown. The percentage of trees in which the associated taxa clustered together is shown next to the branches. Initial tree(s) for the heuristic search were obtained automatically by applying neighbor-join and BioNJ algorithms to a matrix of pairwise distances estimated using the Maximum Composite Likelihood (MCL) approach, and then selecting the topology with superior log likelihood value. The tree is drawn to scale, with branch lengths measured in the number of substitutions per site. The analysis involved 22 nucleotide sequences. There were a total of 1490 positions in the final dataset. Evolutionary analyses were conducted in MEGA7 [24].

The evolutionary history was inferred by using the Maximum Likelihood method based on the Tamura 3-parameter model [20]. The tree with the highest log likelihood (−72,715.28) is shown. The percentage of trees in which the associated taxa clustered together is shown next to the branches. Initial tree(s) for the heuristic search were obtained automatically by applying neighbor-join and BioNJ algorithms to a matrix of pairwise distances estimated using the Maximum Composite Likelihood (MCL) approach, and then selecting the topology with superior log likelihood value. The tree is drawn to scale, with branch lengths measured in the number of substitutions per site. The analysis involved 56 nucleotide sequences. Codon positions included were 1st + 2nd + 3rd + Noncoding. There were a total of 2519 positions in the final dataset. Evolutionary analyses were conducted in MEGA7 [24].

The evolutionary history was inferred by using the Maximum Likelihood method based on the Tamura 3-parameter model [20]. The tree with the highest log likelihood (−74,351.59) is shown. The percentage of trees in which the associated taxa clustered together is shown next to the branches. Initial tree(s) for the heuristic search were obtained automatically by applying neighbor-join and BioNJ algorithms to a matrix of pairwise distances estimated using the Maximum Composite Likelihood (MCL) approach, and then selecting the topology with superior log likelihood value. The tree is drawn to scale, with branch lengths measured in the number of substitutions per site. The analysis involved 32 nucleotide sequences. Codon positions included were 1st + 2nd + 3rd + Noncoding. There were a total of 11998 positions in the final dataset. Evolutionary analyses were conducted in MEGA7 [24]. 

## 4. Discussion

Distribution maps for Ae. species predict the international spread of dengue, Chikungunya, and yellow fever [25]. In Africa, areas of high arbovirus suitability are found within West Africa and parts of Central and East Africa. In these regions, these viruses are not well-distributed in their pattern and occurrence [26]. DENV-1 has been reported in Nigeria (Figure 1) and continues to circulate in many parts of the country. DENV-2 is most predominant in Nigeria with a shift in ecotypes from Sylvatic strains of the past to the importation of Asian 1 lineage (Table 1). Until recently, the DENV-2 genotype II Cosmopolitan strain was first reported in Nigeria (Figure 2). The strain is closely related to isolates from two travelers returning to France from Burkina Faso in Africa [27]. Several reports of various DENV-2 genotypes show a significant deviation from the DENV ecotypes reported in 1964 (Figure 2) when sylvatic strains (Table 1) were identified in Nigeria. The reason for this is agricultural practices which have changed the vegetation cover and also hunting activities that have altered the wildlife population [17]. One DENV-4 sequence was reported during the silent dengue outbreak of 2019 in Edo State in Nigeria. Although the genotype and ecotype could not be ascertained with the dengue virus genotyping tool, it shows a close evolutionary relationship to a strain from Brazil (Figure 3) which also reported the circulation of DENV-2 genotype V Asian I lineage (Figure 2). There was no official report of an ongoing outbreak in the country, but these viruses were detected following increased surveillance and sequencing. Although the latter is not widely performed in most laboratories in Nigeria, there are several reports of continuous detection of all DENV serotypes by PCR. Multiple serotypes circulation within the same geographical space is a major risk factor for DHF, which has been reported to be increasing in Nigeria [6]. DENV-2 and DENV-4 were simultaneously circulating in Benin City, Edo State during the dengue outbreak in 2019 (Figure 2 and Figure 3). This same scenario is typical in parts of Nigeria where more than one DENV serotype circulates [28]. 

The period 1986–1991 was an extraordinarily active period for yellow fever in Africa [2]. The largest number of cases was reported from Nigeria, where the resurgence of yellow fever continues [28]. YFV was identified far back as 1946 in southern Nigeria, and later in Osun State in 1965 and 1969 (Figure 4). Years later, there were two major epicenters of repeated epidemics; one was in Oju during the 1986 sylvatic YF epidemic and Cross-River State in the 1987 epidemic [29]. Over nine villages in the Oju area of Benue State had an overall attack rate of 4.9% with a mortality rate of 2.8% [30]. More YF outbreaks occurred in 2018 and 2019 in Edo State (Figure 4). In 2020, many people in the Ogbadibo and Okpokwu LGAs of Benue State died from YF during the outbreak. Epidemiologists reported that the spread of YF heightened following the refusal by community members to be immunized due to cultural beliefs [28]. In 2021, nine countries in the WHO African Region, namely Cameroon, Chad, the Central African Republic, Côte d’Ivoire, the Democratic Republic of Congo, Ghana, Niger, the Republic of Congo, and Nigeria reported human laboratory-confirmed cases of YF in high-risk areas with a history of YF transmission and outbreaks. As of 20th December 2021, 88 lab-confirmed cases were reported since the beginning of the year. Among the probable cases, there were 66 deaths reported from 6 countries which included Ghana (42 cases), Cameroon (8 cases), Chad (8 cases), Nigeria (4 cases), Congo (2 cases), and the DRC (2 cases), with the overall case fatality ratio (CFR) among the probable cases being 22%. Although some of the affected countries had conflict-affected or vulnerable settings, the situation in Nigeria and other countries was based on low YF population immunity [31]. Delays in the investigation of suspected YF cases due to insecurity in parts of Chad, Cameroon, and the CAR, or under-served (nomadic) communities in Nigeria and Ghana, have implications for harm to human health and the risk of onward amplification and spread [32]. Some major urban areas have reported YF cases and are a major concern, as they pose a significant risk of amplification mediated by *Ae. aegypti* person–mosquito–person transmission (without a sylvatic intermediary). These urban YF outbreaks can rapidly amplify with onward spread internationally, as seen in Angola and the DRC in 2016 [33]. The overall YF vaccination coverage in these regions is not sufficient to provide herd immunity and prevent outbreaks. Estimates from the WHO and UNICEF in 2020 on routine YF vaccination coverage was 44% in the African region, much lower than the 80% threshold required to confer herd immunity against YF. The national coverages in the countries of concern were all under 80%, with the exceptions of Ghana (88%), Congo (69%), Côte d’Ivoire (69%), Niger (67%), Cameroon (57%), the DRC (56%), Nigeria (54%), the CAR (41%), and Chad (35%) [32]. These low YF vaccination coverages indicate the presence of an underlying susceptible population at risk of YF and risk of continued transmission. It is not surprising that outbreaks are occurring in large geographic areas in West and Central Africa, signaling a resurgence and intensified transmission of the YFV in Nigeria inclusive (Figure 4). 

In general, the risk of arboviruses in Nigeria is high due to active virus circulation in areas with a history of past infection, or where vectors and animal reservoirs currently exist [34]. Increased population movement, including vulnerable nomadic populations that are not covered by routine immunization for YF or even dengue (although not yet available in the country) and undocumented border crossings increase the risk of national and regional spread beyond the AFRO region. Clusters of arboviral infections are mixed, occurring in urban and agricultural/forest territories, thereby highlighting the persistent risk of spillover into urban areas (Figure 1, Figure 2, Figure 3 and Figure 4). The affected countries are part of the savannah region with similar ecosystems (forest and shrubland), and a variety of animals, including non-human primates (monkeys), that are the primary wild hosts of the arboviruses [35,36]. The ecosystem is also conducive to the potential vectors involved in the savannah transmission cycle that connects the sylvatic and urban cycles of both humans and primates. 

Although the overall risk at the global level is considered low, as no cases related to the current YF outbreak have been reported outside of the African region, more than 2 billion individuals in Asia live in areas where the competent vectors *Ae. aegypti* and *Ae. albopictus* are present. The expansion of global air travel and the rapid ecological and demographic changes increase the risk of YF introductions into Asia. Based on the interconnectivity with endemic countries, studies have suggested that China, India, the United Arab Emirates, and Saudi Arabia are at the greatest risk of YF introduction; however, the risk of autochthonous transmission is unknown [37]. There is a risk of outbreaks in urban settings, introduced by viremic travelers to largely unprotected urban populations such as Lagos (Nigeria), N’Djamena (Chad), or Bangui (CAR), with a continued risk of rapid amplification internationally. The importation of cases to countries with suboptimal coverage and persisting population immunity gaps pose a high risk to the region and may jeopardize the tremendous efforts invested to achieve elimination. The countries reporting YF cases and outbreaks are all high-priority countries for the Eliminate Yellow Fever Epidemic (EYE) strategy. They have introduced yellow fever vaccination into their routine immunization schedule for those aged 9 months, and also requirements of proof of vaccination against YF for all incoming travelers ≥9 months, except Chad and Nigeria, who request proof of vaccination only for travelers coming from countries with a risk of yellow fever transmission. Vaccination is the primary means for the prevention and control of yellow fever. In urban centers, targeted vector control measures are also helpful to interrupt transmission. The WHO and its partners will continue to support local authorities to implement these interventions to control the current outbreaks [38]. 

The first report of CHIK in Nigeria was in 1963, with sero-surveys showing high antibodies circulating among the populace [39]. In 1969, CHIK was isolated from man and mosquitoes during the first CHIK epidemic and again from children in Ibadan during the second epidemic in 1974 [14]. In the inter-epidemic period from 1970 to 1973, only two strains of the CHIK virus were isolated in Ibadan (Figure 5). Recently, CHIK was reported in Kogi and Kwara States (northcentral Nigeria) [15,35] and Lagos State (southwest Nigeria), showing that it is still a significant problem. 

A limitation in this study is that the DENV-3 sequences from Nigeria could not be verified as they did not fit into the multiple pairwise alignments; hence, the evolutionary history could not be determined.

## 5. Conclusions

These three arthropod-borne viruses are a significant public health problem in Nigeria and continue to infect people across different parts of the country to date. Apart from climatic factors, urbanization, and other socioeconomic factors that define suitable habitats for mosquitoes that transmit these viruses, deforestation has given way to cities where indiscriminate discarding of hollow containers enhances the breeding of these vectors. As a result, arboviral transmission is currently emerging in the West African front. In 2019, due to increased surveillance activities, several outbreaks of dengue and yellow fever were reported in Edo State. In 2021, the NCDC confirmed YF cases, reinforcing the need to strengthen arboviral surveillance around the country, particularly in the tropical rainforest and savannah regions where outbreaks have been reported. With the reports of increased breeding of vectors and molecular details of *Ae. albopictus* and *Ae. aegypti* from Nigeria, activities to destroy breeding places in heavily agrarian communities and urban centers should be put in place and sustained. Experimental infection of *Aedes* species is required to understand the present transmission dynamics in different parts of Nigeria so as to classify locations according to specific arbovirus hotbeds and/or high-risk areas. In dengue-prone areas, the National Primary Health Agency should make efforts to procure dengue vaccines, as they are not presently included in the Expanded Program on Immunization. Prophylaxis with available vaccines in the case of yellow fever and dengue, and symptomatic treatment of already infected persons, are recommended to prevent spread.

## Figures and Tables

**Figure 1 pathogens-11-01162-f001:**
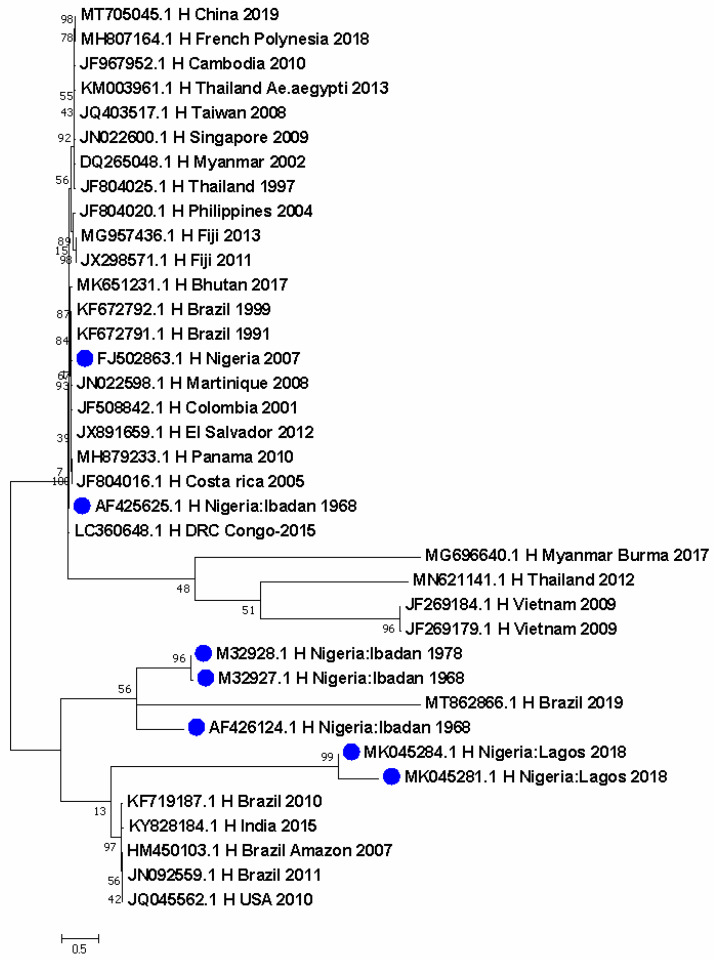
Evolutionary relationship of DENV-1 from Nigeria in comparison with sequences around the world.

**Figure 2 pathogens-11-01162-f002:**
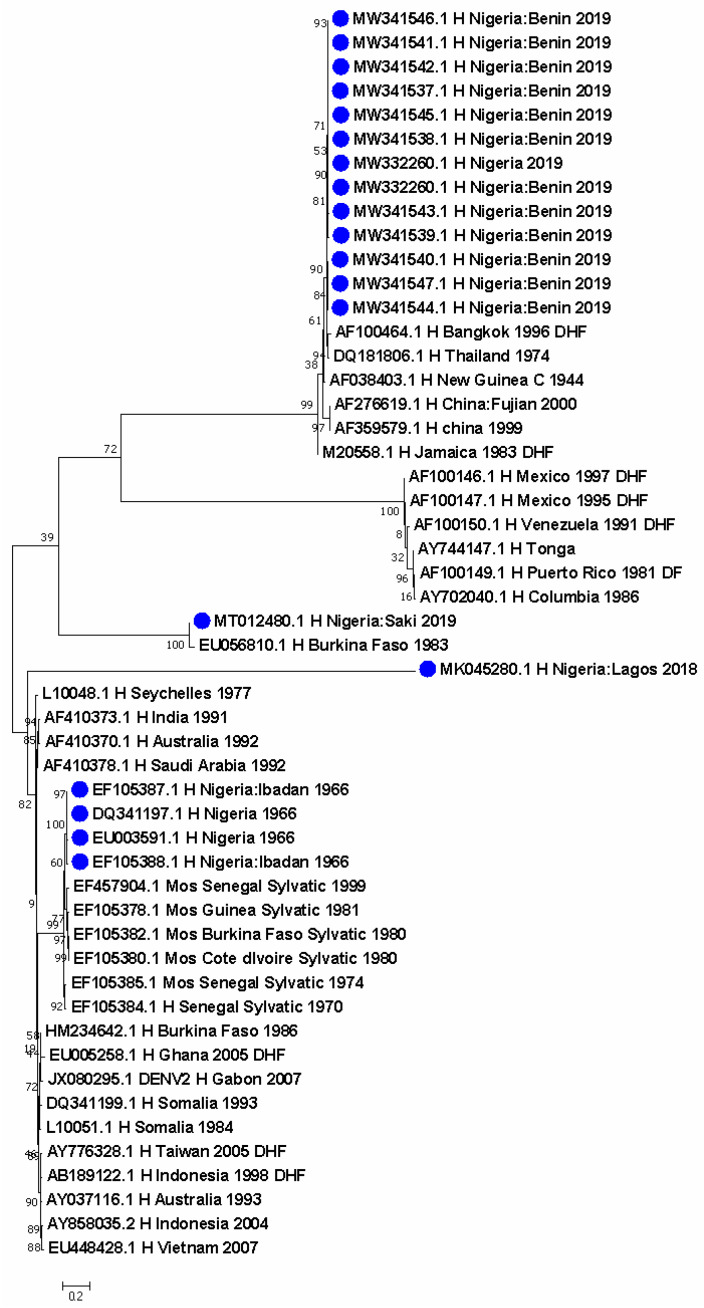
Evolutionary relationship of DENV-2 from Nigeria compared with sequences from around the world.

**Figure 3 pathogens-11-01162-f003:**
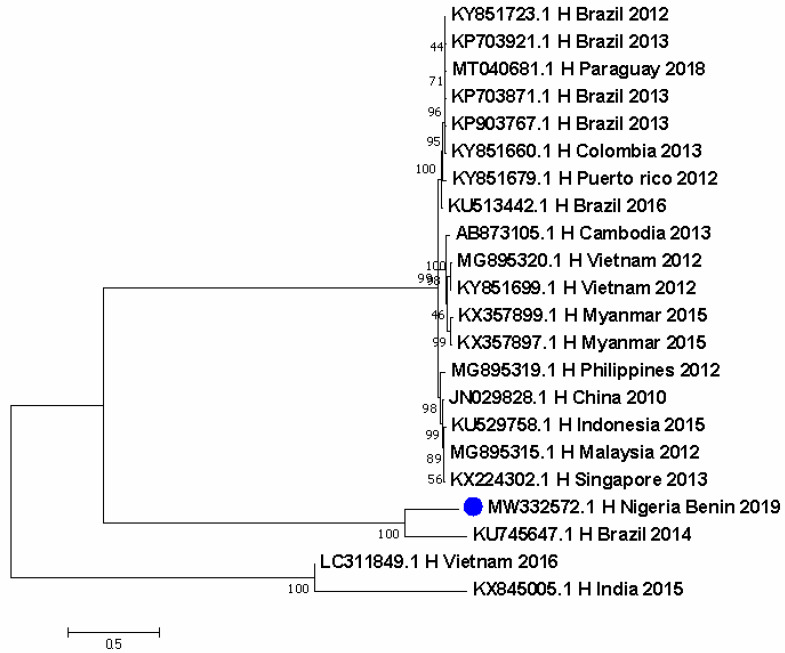
Evolutionary diversity of DENV-4 from Nigeria and other countries.

**Figure 4 pathogens-11-01162-f004:**
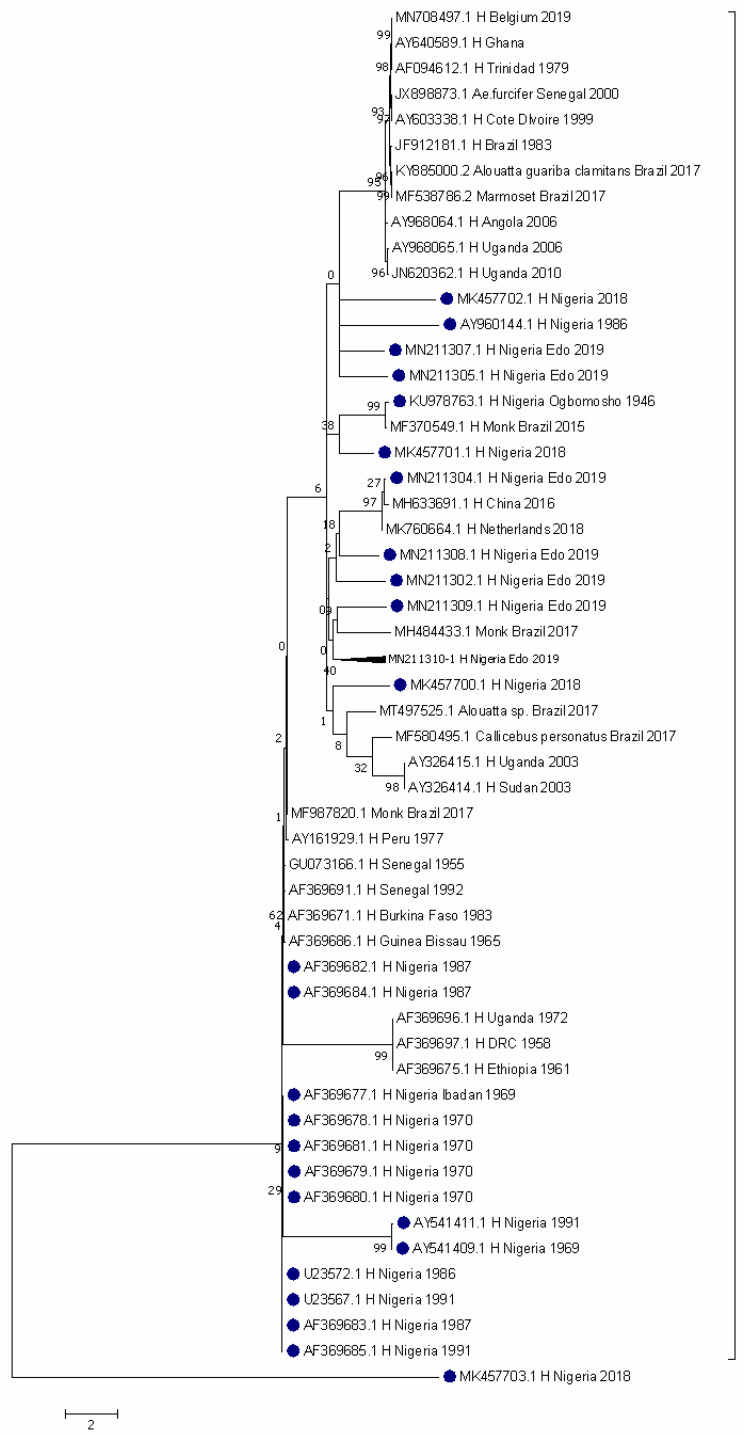
Evolutionary relationship of yellow fever virus sequences in Nigeria and other countries.

**Figure 5 pathogens-11-01162-f005:**
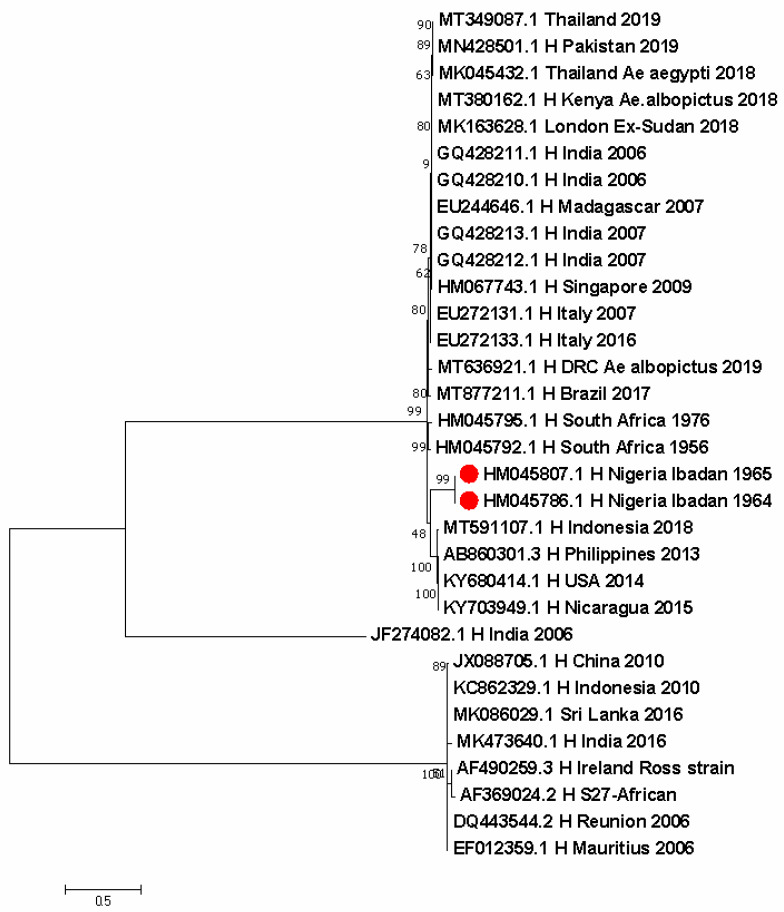
Evolutionary history of Chikungunya virus in Nigeria and other countries.

**Table 1 pathogens-11-01162-t001:** Genotypes, strain, location and source of DENV serotypes in Nigeria from 1966 to 2019.

SN	Serotype	Accession No.	Genotype	Strain	Year	Location	Source
1	DENV-1	M32928.1	Unassigned	Unassigned	1978	Ibadan	Human
2	DENV-1	M32927.1	Unassigned	Unassigned	1965	Ibadan	Human
3	DENV-1	MK045284.1	Unassigned	Unassigned	2018	Lagos	Human
4	DENV-1	MK045281.1	Unassigned	Unassigned	2018	Lagos	Human
5	DENV-1	AF425625.1	Unassigned	Unassigned	1964	Ibadan	Human
6	DENV-1	FJ502863.1	Unassigned	Unassigned	2007	Ibadan	Human
7	DENV-2	EF105387.1	Genotype VI	Sylvatic	1966	Ibadan	Human
8	DENV-2	DQ341197.1	Genotype VI	Sylvatic	1966	Ibadan	Human
9	DENV-2	EF105388.1	Genotype VI	Sylvatic	1966	Ibadan	Human
10	DENV-2	EU003591.1	Genotype VI	Sylvatic	1966	Ibadan	Human
11	DENV-2	MK045282.1	Unassigned	Unassigned	2018	Lagos	Human
12	DENV-2	MK045280.1	Unassigned	Unassigned	2018	Lagos	Human
13	DENV-2	MT012480.1	Genotype II	Cosmopolitan	2019	Saki	Human
14	DENV-2	MW341537.1	Genotype V	Asian I	2019	Benin	Human
15	DENV-2	MW341538.1	Genotype V	Asian I	2019	Benin	Human
16	DENV-2	MW341539.1	Genotype V	Asian I	2019	Benin	Human
17	DENV-2	MW341540.1	Genotype V	Asian I	2019	Benin	Human
18	DENV-2	MW341541.1	Genotype V	Asian I	2019	Benin	Human
19	DENV-2	MW341542.1	Genotype V	Asian I	2019	Benin	Human
20	DENV-2	MW341543.1	Genotype V	Asian I	2019	Benin	Human
21	DENV-2	MW341544.1	Genotype V	Asian I	2019	Benin	Human
22	DENV-2	MW341545.1	Genotype V	Asian I	2019	Benin	Human
23	DENV-2	MW341546.1	Genotype V	Asian I	2019	Benin	Human
24	DENV-2	MW341547.1	Genotype V	Asian I	2019	Benin	Human
25	DENV-2	MW332260.1	Genotype V	Asian I	2019	Benin	Human
26	DENV-3	MK045283.1	Unassigned	Unassigned	2018	Lagos	Human
27	DENV-3	MK045285.1	Unassigned	Unassigned	2018	Lagos	Human
28	DENV-3	MK045282.1	Unassigned	Unassigned	2018	Lagos	Human
29	DENV-4	MW332572.1	Unassigned	Unassigned	2019	Benin	Human

**Table 2 pathogens-11-01162-t002:** Distribution of DENV genotypes, ecotypes in Nigeria from 1964 to 2019.

	DENV-1	DENV-2	DENV-3	DENV-4	Total	*p*-Value
	No. (%)	No. (%)	No. (%)	No. (%)	No. (%)	
**Year**						
1964	1 (3.4)	0 (0.0)	0 (0.0)	0 (0.0)	1 (3.4)	0.02
1965	1 (3.4)	0 (0.0)	0 (0.0)	0 (0.0)	1 (3.4)	
1966	0 (0.0)	4 (13.8)	0 (0.0)	0 (0.0)	4 (13.8)	
1978	1 (3.4)	0 (0.0)	0 (0.0)	0 (0.0)	1 (3.4)	
2007	1 (3.4)	0 (0.0)	0 (0.0)	0 (0.0)	1 (3.4)	
2018	2 (6.9)	2 (6.9)	3 (10.3)	0 (0.0)	7 (24.1)	
2019	0 (0.0)	13 (44.8)	0 (0.0)	1 (3.4)	14 (48.3)	
Total	6 (20.7)	19 (65.5)	3 (10.3)	1 (3.4)	29 (100)	
**Location**						
Ibadan	4 (13.8)	4 (13.8)	0 (0.0)	0 (0.0)	8 (27.6)	0.02
Lagos	2 (6.9)	2 (6.9)	3 (10.3)	0 (0.0)	7 (24.1)	
Saki	0 (0.0)	1 (3.4)	0 (0.0)	0 (0.0)	1 (3.4)	
Benin	0 (0.0)	12 (41.4)	0 (0.0)	1 (3.4)	13 (44.8)	
Total	6 (20.7)	19 (65.5)	3 (10.3)	1 (3.4)	29 (100)	
**Genotype**						
Genotype II	0 (0.0)	1 (3.4)	0 (0.0)	0 (0.0)	1 (3.4)	0.01
Genotype V	0 (0.0)	12 (41.4)	0 (0.0)	0 (0.0)	12 (41.4)	
Genotype VI	0 (0.0)	4 (13.8)	0 (0.0)	0 (0.0)	4 (13.8)	
Unassigned	6 (20.7)	2 (6.9)	3 (10.3)	1 (3.4)	12 (41.4)	
Total	6 (20.7)	19 (65.5)	3 (10.3)	1 (3.4)	29 (100)	
**Ecotype**						
Sylvatic	0 (0.0)	4 (13.8)	0 (0.0)	0 (0.0)	4 (13.8)	0.01
Cosmopolitan	0 (0.0)	1 (3.4)	0 (0.0)	0 (0.0)	1 (3.4)	
Asian	0 (0.0)	12 (41.4)	0 (0.0)	0 (0.0)	12 (41.4)	
Unassigned	6 (20.7)	2 (6.9)	3 (10.3)	1 (3.4)	12 (41.4)	
Total	6 (20.7)	19 (65.5)	3 (10.3)	1 (3.4)	29 (100)	

## Data Availability

All sequences used in the study are publicly available on https://www.ncbi.nlm.nih.gov/ (accessed on 10 December 2020).

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
