# Peer review of "Recurrent Episodes of Some Mosquito-Borne Viral Diseases in Nigeria: A Systematic Review and Meta-Analysis"

_pathogens, 2022, doi:10.3390/pathogens11101162_

Round 1

Reviewer 1 Report

Introduction

1.       Lines 21-24, rewrite sentences 1 and 2 for clarity, chikungunya is an alphavirus.

2.       Lines 26-27, Aedes albopictus may no longer serve as a secondary vector for some viruses

3.       Lines 27-28, rewrite sentence for grammar

4.       Lines 28-31, include references to support information in this sentence

Results:

1.       Line 95, claim in this sentence is unsupported

2.       Tables 1 and 2 legends are insufficient

3.       Figures 1-5 legends are insufficient and incomplete

Discussion:

1.       Lines 128-132, rewrite section for clarity and grammatical flow

2.       Lines 135-140, cite literature to support claims

3.       Lines 160-161, rephrase sentence in line with academic/scientific writing (“due to cultural beliefs” may be more appropriate)

4.       Line 216, correct for grammar

5.       Lines 231-233, should be in a separate paragraph

6.       Line 239, “outbreaks driven by sylvatic strains of YFV”

Author Response

Introduction

  1. Q: Lines 21-24, rewrite sentences 1 and 2 for clarity, chikungunya is an alphavirus.

Author’s response: Done. Now reads “Two of these, yellow fever virus and dengue virus, with Chikungunya which is an alphavirus rank among the most important arthropod-borne diseases in the developing world.”

  1. Q: Lines 26-27, Aedes albopictus may no longer serve as a secondary vector for some viruses

Author’s response: Has been removed.

  1. Q: Lines 27-28, rewrite sentence for grammar

Author’s response: Yellow fever epidemics have been reported in Angola, the Democratic Republic of Congo, Sudan, Nigeria, and across Africa [2-3]. Similarly, dengue has emerged as another important arthropod-borne viral disease in humans in the last 40 years.

  1. Q: Lines 28-31, include references to support information in this sentence

Author’s response: Reference now included.

Results:

  1. Q: Line 95, claim in this sentence is unsupported

Author’s response: Sentence has been properly written to capture the author’s view on DENV-2 in Nigeria.

  1. Q: Tables 1 and 2 legends are insufficient

Author’s response: More content has been added to the legend of Tables 1 and 2.

  1. Q: Figures 1-5 legends are insufficient and incomplete

Author’s response: Full complement of legend has now been included.

Discussion:

  1. Q: Lines 128-132, rewrite section for clarity and grammatical flow

Author’s response. The entire section has been rewritten.

  1. Q: Lines 135-140, cite literature to support claims

Author’s response. Reference cited Liang et al., 2015,

  1. Q: Lines 160-161, rephrase sentence in line with academic/scientific writing (“due to cultural beliefs” may be more appropriate)

Author’s response: Change to due to cultural beliefs

  1. Q: Line 216, correct for grammar

Author’s response: Done.

  1. Q: Lines 231-233, should be in a separate paragraph

Author’s Response: Now in a separate paragraph.

  1. Q: Line 239, “outbreaks driven by sylvatic strains of YFV”

Author’s Response: Rephrased to read “As a result, arboviral transmission is currently emerging in the West African front.”

Reviewer 2 Report

The authors presented here an interesting work, which I consider needs to be revised in order to be published.

- Line 4. Aedes species. “Species” without italics

- Line 21. … genus Flavivirus.

- Line 25. You mean Aedes (St.) aegypti?

- Line 40 to 45. I suggest including the name of the countries as well. It will be easier to understand to readers from other parts of the world.

- Line 45. DRC: Democratic Republic of Congo? Include the complete name of the country the first time you mention it.

- Line 46. … Ibadan, Nigeria, in 1964 …

- Line 54. Aedes species …

- Line 64. “… were retrieved”. Is redundant, in Line 61 you began the sentence mentioning the virus sequences that “were retrieved”. I think that you could reorganize the paragraph.

- Line 66. What “similar viruses” you refer to?

- Line 109. You don’t mention Table 1 in the text.

- Lines 114-123. The results shown in Figures 1 to 5 must be explained in the Results section. The Figures show a result that cannot be mentioned in the Discussion section for the first time.

- Line 128. Is not what you show in Figure 1.

- Line 151. “The worldwide total of …” check English.

- Line 154. I cannot find the 1965 report.

- Line 205. Ae. albopictus (italics)

- Line 208. “…studies have suggested that …”

- Line 247. species (without italics)

- Line 270. Antiviral Res

- Line 278. Check the style for the Journal names.

- Line 305. Journal abbreviation

Author Response

The authors presented here an interesting work, which I consider needs to be revised in order to be published.

Q- Line 4. Aedes species. “Species” without italics

Author’s Response: Done

Q- Line 21. … genus Flavivirus.

Author’s Response: Done

Q- Line 25. You mean Aedes (St.) aegypti?

Author’s Response: Changed

Q- Line 40 to 45. I suggest including the name of the countries as well. It will be easier to understand to readers from other parts of the world.

Author’s Response: Tanzania and Sudan, now included.

Q- Line 45. DRC: Democratic Republic of Congo? Include the complete name of the country the first time you mention it.

Author’s Response: This has been done with DRC in bracket the first time used.

Q- Line 46. … Ibadan, Nigeria, in 1964 …

Author’s Response: Ibadan Nigeria in 1974 has now been included.

Q- Line 54. Aedes species …

Author’s Response: Aedes has been italicized

Q- Line 64. “… were retrieved”. Is redundant, in Line 61 you began the sentence mentioning the virus sequences that “were retrieved”. I think that you could reorganize the paragraph.

Author’s Response: were retrieved has been removed. Paragraph now reads ‘The search words in the nucleotides and genome database were dengue virus, yellow fever virus, Chikungunya, and Nigeria.’’ 

Q- Line 66. What “similar viruses” you refer to?

Author’s Response: Has been changed to these three viruses

Q- Line 109. You don’t mention Table 1 in the text.

Author’s Response: Done. Dengue is highly endemic in Nigeria, with intermittent outbreaks of Asian lineage in Edo State (Table 1).

Q- Lines 114-123. The results shown in Figures 1 to 5 must be explained in the Results section. The Figures show a result that cannot be mentioned in the Discussion section for the first time.

Author’s response: Figures 1-5 have been explained in results section now.

Q- Line 128. Is not what you show in Figure 1.

Q- Line 151. “The worldwide total of …” check English.

Author’s Response: Thank you. Modified and now reads, the period 1986-1991 was an extraordinarily active period for yellow fever in Africa [2]. The largest number of cases was reported from Nigeria, where the resurgence of yellow fever continues [28].

Q- Line 154. I cannot find the 1965 report.

Author’s Response: It has been modified to read. YFV was identified far back as 1946 in southern Nigeria, and later in Osun State in 1965 and 1969 (Figure 4)

Q- Line 205. Ae. albopictus (italics)

Author’s Response: Modified

Q- Line 208. “…studies have suggested that …”

Author’s Response: Modified

Q- Line 247. species (without italics)

Author’s Response: Italics removed

Q- Line 270. Antiviral Res

Author’s Response: Full stop added

Q- Line 278. Check the style for the Journal names.

Author’s Response: Checked all journal names and corrected accordingly

Q- Line 305. Journal abbreviation

Author’s Response: All the journal abbreviations have been properly captured.

Reviewer 3 Report

Figures 2 and 4 are not clearly visible; please enhance their quality as in figures 1 and 3. Please include a concise description for tables 1 and table 2 under the caption that corresponds to each table. It provides the readers with a basic summary of the table's purpose to assist them to comprehend it. Since Table 2 hasn't been covered in the discussion section, please add a brief paragraph under the title "discussion" summarizing Table 2's conclusion. In the result section, only the result from table 2 has been mentioned, thus cite table 1 and figures 1, 2, 3, 4, and 5. Please add a final heading, "Conclusion," and provide a conclusion to the entire study.

Author Response

Q: Figures 2 and 4 are not clearly visible; please enhance their quality as in figures 1 and 3.

Author’s Response: Maximum Likelihood method was used to draw the trees again with adjustments for better visibility.

Q: Please include a concise description for tables 1 and table 2 under the caption that corresponds to each table. It provides the readers with a basic summary of the table's purpose to assist them to comprehend it.

Author’s response: This has been done.

Q: In the result section, only the result from table 2 has been mentioned, thus cite table 1 and figures 1, 2, 3, 4, and 5.

Author’s Response: Results for Fig. 1-5 now added.

Q: Please add a final heading, "Conclusion," and provide a conclusion to the entire study.

Author’s Response: The conclusion section has been added, with a summary of the study  

Round 2

Reviewer 2 Report

Please, check Line 362: it should say "Aedes species"